# Rapid On-Site Phenotyping via Field Fluorimeter Detects Differences in Photosynthetic Performance in a Hybrid—Parent Barley Germplasm Set

**DOI:** 10.3390/s20051486

**Published:** 2020-03-08

**Authors:** Miriam Fernández-Calleja, Arantxa Monteagudo, Ana M. Casas, Christophe Boutin, Pierre A. Pin, Fermín Morales, Ernesto Igartua

**Affiliations:** 1Department of Genetics and Plant Production, Aula Dei Experimental Station (EEAD-CSIC), Avda. Montañana 1005, E-50059 Zaragoza, Spain; mfernandez@eead.csic.es (M.F.-C.); amonteagudo@eead.csic.es (A.M.); acasas@eead.csic.es (A.M.C.); 2Syngenta Seeds SAS, 12 Chemin de l’Hobit, 31790 Saint Sauveur, France; christophe.boutin@syngenta.com (C.B.); pierre.pin@secobra.com (P.A.P.); 3Agrobiotechnology Institute (IdAB), CSIC-Gobierno de Navarra, Avda. de Pamplona 123, 31192 Mutilva, Navarra, Spain; fmorales@eead.csic.es

**Keywords:** chlorophyll fluorescence, photoprotection, proximal sensing, hybrid breeding, drought stress

## Abstract

Crop productivity can be expressed as the product of the amount of radiation intercepted, radiation use efficiency and harvest index. Genetic variation for components of radiation use efficiency has rarely been explored due to the lack of appropriate equipment to determine parameters at the scale needed in plant breeding. On the other hand, responses of the photosynthetic apparatus to environmental conditions have not been extensively investigated under field conditions, due to the challenges posed by the fluctuating environmental conditions. This study applies a rapid, low-cost, and reliable high-throughput phenotyping tool to explore genotypic variation for photosynthetic performance of a set of hybrid barleys and their parents under mild water-stress and unstressed field conditions. We found differences among the genotypic sets that are relevant for plant breeders and geneticists. Hybrids showed lower leaf temperature differential and higher non-photochemical quenching, resembling closer the male parents. The combination of traits detected in hybrids seems favorable, and could indicate improved photoprotection and better fitness under stress conditions. Additionally, we proved the potential of a low-cost, field-based phenotyping equipment to be used routinely in barley breeding programs for early screening for stress tolerance.

## 1. Introduction

Plant breeding has experienced an explosion of advances in the last 30 years, with the development and release of molecular, genomic, bioinformatic and technological tools and resources that accelerate crop improvement [1]. Not all areas developed, however, at the same pace. Currently, phenotyping represents the main bottleneck for effective selection of interesting genotypes. This is true for all breeding methods, but especially for the increasingly used genomic selection, for which phenotyping is key to train prediction models [2]. When following a genomic selection strategy, precision-phenotyping of the training population is most important, because that dataset provides the basis for developing the statistical model that is then used to predict phenotypic performance in related members of a breeding population [3]. The rise of high-throughput phenotyping platforms (HTPPs) has enabled large-scale, rapid and accurate data collection under controlled and field conditions [4,5], but their impact on crop improvement has yet to be fully realized [6]. To exploit the potential of high-throughput phenotyping for crop breeding, a direct phenotypic evaluation under adequately monitored heterogeneous real field conditions must be performed [7]. Also, more flexible and affordable approaches need to be developed [8], combining user-friendly data management tools and open-source platforms. 

Recent advances in high-throughput phenotyping, particularly field-based, have boosted the power of trait-based crop breeding (“physiological breeding”) [9]. Physiological breeding involves the design of an improved plant type, the cross of parents with different complex but complementary traits to achieve cumulative gene action for yield, and the use of phenomic and genomic information to select the best progeny. An example of successful implementation of high-throughput phenotyping on physiological breeding comes from CIMMYT. Screening of genetic resources for spectral indices associated with temperature, water content, and pigment composition of leaves via thermal and multispectral imagery was employed to identify complementary parental sources for adaptive traits. This approach allowed to deliver a new generation of drought adapted lines built by pyramiding strategic combinations of stress adaptive traits [10]. Similarly, high-throughput phenotyping can be used to screen genetic resources for traits related to radiation use efficiency, photosynthesis, and crop biomass, which build crop productivity [11], alternatively to the already exploited harvest index [12], and thus explore new traits that are usually beyond the reach of plant geneticists and breeders.

The PhotosynQ app (https://photosynq.org), developed in the laboratory of David Kramer (Michigan State University, Michigan, USA), is an open access collaborative platform that allows the plant research community to collect, analyze, discuss and share plant photosynthetic-related data using a low-cost handheld device, the MultispeQ [13]. This instrument combines a pulse-amplitude-modulated fluorimeter, a chlorophyll meter, and a spectrometer into one. Therefore, it provides valuable information about crop status and photosynthetic performance in a single reading in a time scale of seconds. It provides indirect measurements of a number of parameters, which have been benchmarked against standard laboratory devices [13].

Traditionally, leaf photosynthetic properties have been measured using slow and laborious gas exchange methods, unfit for testing the large populations managed in breeding programs. Chlorophyll fluorescence is a fast, non-destructive method aimed to investigate the efficiency of the photosystem II (PSII) of plants. The light energy absorbed by leaf chlorophylls and carotenoids can undergo three competing fates: (1) it can be used to drive photosynthesis (photochemistry), (2) the excess energy can be dissipated as heat, or (3) it can be re-emitted as chlorophyll fluorescence [14] (Figure 1). Monitoring the dynamics of chlorophyll fluorescence in response to beam pulses allows measuring these three complementary energy pathways and, estimating the amount of energy from photosystem II which goes to photochemistry (φ_II_), is dissipated as heat (monitored as non-photochemical quenching) (φ_NPQ_), and is dissipated in a non-regulated way (φ_NO_) [15]. Chlorophyll fluorescence-based methods require prior dark adaptation of the plant leaves to be measured for full characterization of the chlorophyll fluorescence yields and associated parameters. For some purposes, it is possible to use chlorophyll fluorescence measurements of light-adapted leaves, which considerably reduces the sampling time, allowing large-scale use in the field [16,17]. In the case of MultispeQ, the latter approach is achieved by reproducing the ambient photosynthetically active radiation (PAR) intensity inside the leaf chamber.

Chlorophyll fluorescence measurements can provide information on the status of the photosynthetic apparatus in response to various environmental factors, allowing the early detection of abiotic stresses before the appearance of visible symptoms [18,19,20]. Within abiotic stresses, drought is the most limiting factor for the global crop production in arid and semi-arid areas [21], causing the highest yield losses all over the world [22], and with increasing frequency in the Mediterranean region [23]. To date, numerous QTLs controlling drought tolerance-related traits have been mapped and many attempts have been made to use these major QTLs in water stress-tolerant crop development. However, accurate high-throughput phenotyping, especially under field conditions, is still a limiting factor for breeding of water stress-tolerant varieties [24].

In this context, this study applies a rapid, low-cost and, reliable high-throughput phenotyping tool to explore genotypic variation for stress tolerance of barley under fluctuating field conditions. This is performed in a hybrid-parent barley germplasm set. Hybrid cultivars are used in many crops in which heterosis produces significant yield increases. Hybrids in autogamous crops, like barley, can be developed when male-sterility systems are available. This was true for barley only recently [25]. For this reason, hybrid barley has received increasing attention as a way to increase productivity per unit area, due to its greater yield potential and yield stability compared to conventional varieties, especially under stress conditions [25,26,27]. The objective of this study was to examine potential differences in radiation use efficiency and photosynthetic-related traits between parents (inbred lines) and hybrids under water-stressed and unstressed field conditions, which could indicate environmental adaptation features. Additionally, we wanted to test the potential of a low-cost, field-based phenotyping equipment to be used routinely in barley breeding programs for early screening for stress tolerance. 

## 2. Materials and Methods

### 2.1. Plant Material and Experimental Design

The field trial was carried out during the 2017/2018 crop season in the Aula Dei Experimental Station, of the Spanish National Research Council (EEAD-CSIC), Spain (ED50/UTM zone 30N: X 682163, Y 4622293). The climate conditions of the region represent a continental Mediterranean climate, with a mean, maximum and minimum daily air temperature of 12.9, 19.6, and 6.8 °C, respectively, average relative humidity of 70.0%, mean solar irradiance of 187.2 W m^−2^, and accumulated precipitation of 335 mm during the season. The soil type was silt-loam, composed of 24% clay, 53% silt and 23% sand. 

The barley (*Hordeum vulgare* L.) germplasm set used in this study comprised two female parents, 20 pollinators and 39 hybrids, derived from the cross of the other two (1 hybrid failed). The female parents are cytoplasmic male sterile (CMS) elite inbred lines used in the development of 6-row winter barley hybrids for Europe by Syngenta®. The pollinators are advanced inbred lines developed in the framework of the Spanish Barley Breeding Program [28], well adapted to the Mediterranean conditions, and without fertility restorer genes. The resultant offspring are male-sterile hybrids (F_1_F), an intermediate step in the production of a three-way hybrid, after crossed with a restorer genotype (with the capacity to restore fertility).

Genotypes were sown on November 8th 2017 in 2.4 m^2^ unreplicated plots (4-row, 0.2 m apart, 3 m long). The field trial consisted of 62 plots distributed in four rows of 20 entries. The analyses were performed on the two sets of parents, and the two sets of hybrids (split according to the female parent). Although each genotype was replicated only once, we considered genotypes in each set (female parents, male parents, hybrids) as replicates. The field plot was flood irrigated on September. For fertilization, 500 kg ha^−1^ of base dressing was applied on October (12-24-8) and 450 kg ha^−1^ of top-dressing was applied on March (urea 46%).

### 2.2. Samplings 

Pulse-amplitude modulation (PAM) fluorimetry, chlorophyll content, and leaf temperature differential (LTD) measurements were carried out using a MultispeQ v1.0 device controlled by the PhotosynQ platform software [13].

Measurements were taken in two relevant moments for the crop physiology, after a period of water stress (5 April 2018), and after an episode of abundant rain (24 April 2018). Table 1 shows a summary of climatic variables that characterize the two sampling dates. The first sampling, hereinafter “water-stressed”, was characterized by a lower precipitation accumulated prior the sampling, whereas the second sampling date, hereinafter “unstressed”, was characterized by a higher precipitation accumulated prior the sampling date and higher irradiance and mean temperature during the sampling date. During the water-stressed sampling, all plots were at the growth stage of stem elongation (Z34–Z37 [29]), and during the unstressed sampling plants were at booting stage (Z39–Z47). Samplings were performed on clear days, during day central hours around solar midday. Two MultispeQ devices were used to reduce the sampling time. Measurements were taken on four horizontal upper-most fully expanded leaves per plot, two with each device. We designed a route of genotype sampling across the trial to avoid time and light intensity effects affecting differentially to parents and hybrids. The route drove us across the trial, interspersing the upper and bottom rows, as well as the different genotypic sets. Moreover, to reduce the error derived of the under-representation of the female set compared to the hybrid and male set, we measured each female plot several times (32 data points per plot), intercalated in time between the rest of the genotypes.

Barley leaves did not completely cover the light guide. Therefore, following the developers’ protocol (https://help.photosynq.org/), we built a mask for the light guide in order to reduce the aperture of measurements (Appendix A). After this adjustment, both devices were recalibrated for chlorophyll content measurements. The “MultispeQ v1.0 Leaf Photosynthesis (Masked)” protocol (https://photosynq.org/) was run to measure environmental variables, chlorophyll fluorescence yield changes, and light-induced absorbance changes. The associated macro was applied to calculate photosynthetic efficiency and crop status-related parameters based on the measurements taken.

### 2.3. Chlorophyll Fluorescence, Absorbance and Environmental Variables Measurements

Using the “MultispeQ v1.0 Leaf Photosynthesis (Masked)” protocol, the measurement automatically started once the leaf was clamped. A single leaf measurement took about 8 s, and twice that time to complete the whole operation until clamping the next plant. The protocol provided specific light emitting commands and measurement instructions to the device, obtaining the following parameters: 

Firstly, saturation pulse chlorophyll fluorescence yield parameters (F_s_, F_m_′ and F_0_’) in light-adapted leaves were measured. Steady-state fluorescence yield, F_s_, was collected under continuous actinic light. Following the F_s_ measurement, the sample was exposed to a brief saturating light pulse to obtain an estimate of the maximal fluorescence yield under steady-state illumination, F_m_′, with steady-state levels of non-photochemical quenching (NPQ), but with all PSII centers closed. Immediately after the saturation pulse, the actinic light was switched off. A pulse of far-red light, from a LED emitting at 730  nm, was applied to fully oxidize the plastoquinone pool and Q_A_, allowing measurement of F_0_′ in the presence of steady-state levels of NPQ, but with all PSII centers oxidized [30]. Secondly, transmittance through the leaf of red light (650 nm, chlorophyll absorbed) and infrared radiation (940 nm, non-chlorophyll absorbed), relative to a blank (ambient air) was recorded. Finally, ambient light intensity (PAR) in µmol photons m^−2^ s^−1^, ambient temperature (Ta) and leaf temperature (Tc) in °C were recorded. 

### 2.4. Photosynthetic and Physiological Parameter Calculations

The associated macro used the above-mentioned measurements to automatically calculate the following variables. The yield of variable fluorescence in the light (F_v_’) was calculated as F_m_’−F_0_’, from which the intrinsic efficiency of PSII in the light (F_v_′/F_m_′) was determined. The actual PSII efficiency (φ_II_ ) was estimated as the ratio (F_m_’−F_s_)/F_m_’ [31]. 

The coefficient for photochemical quenching, qP, which relates to the fraction of PSII centers that are “open” based on the “puddle” model of PSII, was calculated as (F_m_’−F_s_)/(F_m_’−F_0_’) [32]. The fraction of PSII centers that are “open” based on the “lake” model of PSII (qL) according to [15] was estimated as (F_m_’−F_s_)/(F_m_’−F_0_’) × (F_0_′/F_s_). The fluorescence decline ratio in steady-state conditions (RFd) was determined as (F_m_’−F_s_)/F_s_ [33]. 

The quenching due to non-photochemical dissipation of absorbed light energy (NPQ_t_) was calculated according to [34], who assumed that the maximal quantum efficiency observed in a range of plants was about 0.83 [35], allowing the calculation of NPQ_t_ without the use of fluorescence yield parameters that require dark acclimation. The yield induced by downregulatory processes (φ_NPQ_) and the yield for other energy losses (φ_NO_) were calculated applying the equations derived by Kramer et al. (Equations 51 and 52) [15] as modified by Tietz et al. [34]. The three light-adapted parameters add up to unity (φ_II_ + φ_NPQ_ + φ_NO_ = 1) [15].

The linear electron flux (LEF) was estimated by multiplying φ_II_ × incident PAR × 0.5 (assuming an equal distribution of excitation between photosystems II and I), and × 0.84, which is considered the most common leaf absorbance coefficient for healthy C_3_ plants [36]. 

Relative chlorophyll content (RC) was calculated as 100 × log (transmittance@940/transmittance@650), which is expressed per unit area and it is the value integrating the chlorophyll content per unit mass and leaf thickness [13]. Leaf temperature differential (LTD) was determined as Tc-Ta.

### 2.5. Statistical Analysis

For the set of genotypes sampled under both water conditions (39 F_1_F, two females, and 20 male parents), differences in RC, LTD, φ_II_, φ_NO_, φ_NPQ_, NPQ_t_, RFd, qL, qP, F_m_’, F_0_’, F_s_, LEF, and F_v_’/F_m_’ between genotypes, environments (unstressed vs. water-stressed conditions), and devices (MultispeQ instruments) were evaluated using the analysis of variance (ANOVA) procedure in R [37]. The ANOVA model included *genotype*, *environment*, *device*, square root of PAR (*sqrt (PAR)*) and *genotype by environment* interaction. Female parents (n = 2), male parents (n = 20), hybrids from female A (n = 19), and hybrids from female B (n = 20) were defined as four genotypic sets, representing four categories potentially having different genetic features. The *genotype* term was subdivided into two components, *genotypic sets* and *genotypes within genotypic sets*. *Genotype* and its subdivision, *environment* and *device* were all considered fixed factors, whereas *sqrt(PAR)* was used as covariate. The light intensity variable was transformed using a √x function, to make its effect on photosynthetic parameters linear. The intraplot variability (measurements on 4 leaves per plot) was considered as error for the main terms (*genotypes within genotype sets*, *environment*, and *device*). The differences between *genotypic sets*, the main objective of this study, were tested against the mean square error of *genotypes within genotypic sets*, because these were independent measurements of these factors. *Genotypic sets* were broken down into the most informative contrasts: hybrids vs. parents, hybrids vs. females, hybrids vs. males, females vs. males, and finally, hybrids from female A vs. hybrids from female B. The interactions of all these contrasts with *environment* were also tested. Means were compared using least significant difference (LSD) test (P < 0.05).

## 3. Results

There was a significant effect of water stress on all the variables assessed, except for F_s_ and φ_NPQ_, as summarized in Table 2. Differences among *genotypic sets* (female parents, male parents and hybrids) were significant for RC, LTD, φ_II_, φ_NO_, φ_NPQ_, NPQ_t_, qL, qP, LEF, RFd and F_v_’/F_m_’. The analysis showed few and weak interactions between the *environment* and the genotypic set. 

Despite having applied the masks to the light guides and having recalibrated both devices with the same calibration cards, the *device* factor had an effect on most variables. However, as the replicates measured with each device were balanced (2 plants per plot measured with each device), we trust the *environment* and *genotype* effects shown by the ANOVA. 

The ANOVAs were carried out with the raw data, even though some deviations from normality and heterogeneity of variances were detected (Appendix A). Data transformation did not improve these results. Deviations from normality usually do not affect the result of the analysis for reasonable sample sizes, and the same is true for violations of the assumption of homogeneity of variances, which is less critical when the sample sizes are substantially different [38].

### 3.1. Chlorophyll Fluorescence-Derived Parameters 

Saturation pulse chlorophyll fluorescence yield parameters were significantly influenced by *environment* (Table 2). We observed an increase in the minimal fluorescence level in the light adapted state, F_0_′, under drought stressed conditions compared to unstressed conditions. In the same direction, F_m_’, which measures the maximal fluorescence level under steady-state illumination, increased in plants under water stress. From the latter parameter, the intrinsic PSII efficiency in the light (F_v_′/F_m_′) was determined. Measurements under unstressed conditions showed increased F_v_′/F_m_′. *Environment* had no influence in F_s_ steady-state fluorescence yield. However, the fluorescence decline ratio in steady-state conditions (RFd) was higher under drought stress conditions. 

Regarding quenching parameters, both photochemical quenching, estimated through “puddle” and “lake” models (qP and qL, respectively), and non-photochemical quenching (NPQ_t_, in light-adapted leaves) were higher under drought stress conditions. 

No differences were found in direct PAM fluorimetry measurements (F_s_, F_m_′ and F_0_’) between *genotypic sets*. However, contrasts among genotypic groups were significant for F_v_′/F_m_′ and RFd. The intrinsic PSII efficiency in the light was higher in the female parents, compared to male parents and hybrids. RFd was higher in both parents than in their progeny. The photochemical quenching coefficients qP and qL, which represent the fraction of open PSII centers, had similar values for the three *genotypic sets*, whereas the non-photochemical quenching parameter NPQ_t_ showed the largest value in hybrids, the lowest in female parents, and an intermediate value in male parents (Table 3). 

### 3.2. Absorbance-Based Parameters

Drought stress had a significant effect on relative chlorophyll content. RC mean value recorded under water-stressed conditions was 20 units lower than the average of non-stressed plants. Female parents showed higher RC values than male parents and hybrids (Table 3).

### 3.3. PSII Energy-Absorbed Allocation Proportions

For all genotypes, the analysis of the partition of absorbed excitation energy in PSII, under both water conditions, showed that 30% of the flux of excitation energy was allocated towards the photochemical (φ_II_) pathway, whereas 70% was devoted to non-photochemical pathways (φ_NPQ_ + φ_NO_). We found differences in the energy allocation ratios, both between *environments* and *genotypes* (Table 2). Higher values of the actual PSII efficiency, φ_II_, were obtained under water-stressed conditions (0.321) compared to the unstressed sampling (0.270). A lower quantum yield of non-regulated energy dissipation of PSII (φ_NO_) was found under water-stressed conditions, with a mean value of 0.24, while non-stressed plants averaged 0.27. There was no *environment* effect on the quantum yield of regulated energy dissipation of PSII (φ_NPQ_) (water-stressed = 0.44, unstressed = 0.46) (Table 3).

Regarding genotypic differences, both types of parents showed higher φ_II_ than hybrids. φ_NO_ was higher in female parents than in male parents and hybrids; and hybrids showed the highest φ_NPQ_, followed by the male parents, while the lowest value was recorded for the female parents (Figure 2). 

The LEF was higher under water-stressed conditions, which correlates with the larger qP under drought stress conditions. Contrasting *genotypic sets*, hybrids and male parents showed a larger electron transport rate than female parents (Table 3), even though the female parents showed the highest φ_II_ value.

### 3.4. Leaf Temperature Differential

The difference between canopy and air temperatures in water-stressed leaves was 1.44 °C lower than those leaves sampled under unstressed conditions (Table 3). Hybrids and male parents showed a different LTD dynamic in contrast to the female parents (Figure 3). No LTD differences were found among genotypic groups under water-stressed conditions. Under unstressed conditions, however, the leaves of male parents and hybrids were significantly cooler than those of the female parents.

### 3.5. Relationship between Chlorophyll Fluorescence-Based Parameters and Crop Status Indicators

The correlation between leaf temperature differential and chlorophyll fluorescence-based parameters was low for the measurements carried out under unstressed plants. However, under drought stress, we found a moderate correlation between LTD, a common indicator of plant stress, and quantum yields of PSII (Figure 4). 

There was no apparent relationship between φ_II_ and LTD, neither under unstressed conditions, nor under water-stressed conditions (r_unstressed_ = −0.02, r_stressed_ = −0.02) (Figure 4a). On the other hand, non-photochemical energy loss quantum yields showed moderate correlation with LTD, being this stronger under the drought stress environment. Φ_NO_ presented a low negative correlation with LTD for unstressed plants (r_unstressed_ = −0.10), whereas in the water-stressed sampling, a positive correlation was found (r_stressed_ = 0.30), meaning that higher quantum yields of non-regulated energy loss correlated with lower differences between canopy temperature and air temperature (Figure 4b). We found a low positive correlation between Φ_NPQ_ and LTD under unstressed conditions (r_unstressed_ = 0.12), while under stressed conditions, the sign of this correlation was reversed (r_stressed_ = −0.16) (Figure 4c).

## 4. Discussion

Improving photosynthesis efficiency is a potential strategy for increasing crop yields. Nevertheless, this will only be achievable if available genetic variation for this trait exists in crop germplasm resources. Chlorophyll fluorescence has been routinely used for many years to monitor the photosynthetic performance of plants in a fast, non-invasive way [14], although certainly its most interesting advantage is the early detection of stress responses [39,40]. In particular, fluorescence can give insights into the ability of a plant to tolerate environmental stresses and into the extent to which those stresses have damaged the photosynthetic apparatus [20]. However, measurements of chlorophyll fluorescence in the field are scarce [41,42,43] compared to those reported under controlled conditions, and also the information derived from them has not been exploited by breeding programs [44]. Moreover, there is a lack of adequate and affordable equipment to carry out rapidly this kind of measurements under field conditions [8].

### 4.1. Drought Stress Indicators

Plants that accumulated a precipitation of just 9.6 mm during the three weeks prior to the sampling day (April 5) experienced more water stress than after receiving 90.2 mm (April 24). This was reflected in a higher leaf temperature (lower LTD) in water-stressed than in unstressed plants, suggesting stomatal closure. Stomatal closure, monitored through stomatal conductance, is considered as a reference parameter of the water stress at which plants are exposed [45]. 

Drought stress had an important effect on photosynthetic efficiency, energy allocation, and crop status as indicated by the measurements derived from the MultispeQ device. Surprisingly, the comparison of the radiation partitioning between *environments*, pointed at a higher proportion of the incident radiation allocated towards photochemistry (Φ_II_) under water-stressed conditions. This finding was paired with the observation of a higher proportion of the energy allocated towards heat dissipation (Φ_NPQ_) and other non-regulatory processes (Φ_NO_) in the unstressed environment. Commonly, Φ_II_ declines with drought stress in studies involving chlorophyll fluorescence measurements and water-stress, [46,47] under controlled conditions. However, other authors have reported the increase of Φ_II_ in water-stressed plants, which was related to higher photorespiration rates [19,48]. This situation occurs commonly when the intensity of water stress is mild or moderate (see [49] for a review). Seemingly, the higher rate of photorespiration in plants during drought stress removed the electron pressure and allowed PSII to work at a higher efficiency. Higher values of Φ_II_ have been correlated with a higher proportion of open reaction centers [50], consistent with our results. Changes in qP are due to closure of reaction centers, resulting from a saturation of photosynthesis by light [20]. Therefore, the higher values of qP and qL found under water stress could be a response to lower light intensity during the sampling (see Table 1). Despite showing a higher Φ_II_, the drought-stressed leaves developed a stronger non-photochemical quenching, expressed as NPQ_t_, than those leaves from unstressed plants, agreeing with other studies [19]. This effect was not seen for Φ_NPQ_. Indeed, the efficiency of the open PSII reaction center, estimated by F_v_′/F_m_′, was higher in the unstressed plants, indicating less heat dissipation than in the water-stressed plants [51]. Φ_NO_ was higher in the unstressed conditions, indicating higher proportion of the energy absorbed by PSII lost in a non-regulated way. The chlorophyll fluorescence decrease ratio Rfd was higher under drought stress conditions, contrary to the results reported by Yao et al. [46]. Rfd is an indicator of the potential photosynthetic activity [52] and, when measured at saturation irradiance, has been directly correlated to the net CO_2_ assimilation rate of leaves [53]. In the same line, LEF was higher under water-stressed conditions. Assuming that a constant proportion of the reductants resulting from LEF is utilized for CO_2_ assimilation, then the PSII operating efficiency would be predicted to be directly proportional to the operating quantum efficiency of CO_2_ assimilation [31,54].

Regarding crop status measurements, relative chlorophyll content decreased under drought stress conditions, which is in agreement with previous results [55,56]. The ability of the plant to maintain high chlorophyll contents, or stay-green, has been associated with improved yield and transpiration efficiency under water-limited conditions [57,58]. Moreover, LTD was lower in absolute terms in water-stressed plants. These values correlate favorably with [59] and further support the close relationship between leaf temperature increases and stomatal closure due to water stress.

Under stress conditions, we found low but still significant correlations between LTD and quantum yields of PSII Φ_NPQ_ and Φ_NO_. The mild stress level observed could be the reason for the relatively low correlations between LTD and the fluorescence parameters. The relationship between Φ_NPQ_ and LTD was negative, pointing at a better crop status (fresher leaves) associated with a higher proportion of the energy absorbed by PSII dissipated in a regulated way. On the other hand, the relationship between Φ_NO_ and LTD was positive, indicating that plants that showed a better crop status (fresher leaves) also showed a lower proportion of the energy absorbed by PSII dissipated in a non-regulated way. Considering the latter relationships found under water stress conditions, and that leaf temperature has been accepted as a crop water stress indicator for a long time [60], we propose to investigate further a possible role of Φ_NPQ_ and Φ_NO_ as potential proxies for breeding for water stress tolerance. 

### 4.2. Photoprotective Response of Hybrids and Fitness under Unfavorable Conditions

We detected genotypic variability in photosynthetic traits. The intrinsic efficiency of PSII was higher in the female parents, compared to male parents and hybrids, pointed at a lesser heat dissipation in the female parents. RFd was higher in both parent sets than in their progeny, what might be indicating a lower net CO_2_ assimilation rate of leaves in the hybrids [53]. However, the non-photochemical quenching parameter NPQ_t_ showed the largest value in hybrids, hinting at a higher dissipation of ‘excess’ light energy absorbed by light-harvesting complexes as heat, harmlessly, preventing the accumulation of reactive intermediates. This observation is supported by the fact that dissipation of excess excitation energy at the level of the PSII antennae has been proved to be the major protective mechanism against the deleterious effects of high light in dehydrating leaves [48]. 

The analysis of the quantum yields of PSII across genotypes showed a slightly higher PSII operating efficiency in the parents, while hybrids stood out by its high Φ_NPQ_ value. Moreover, female parents devoted a clearly higher proportion of the energy absorbed by PSII to other non-regulated energy dissipation processes. It has been reported that at high quantum flux densities, when Φ_II_ values approach zero, high values of Φ_NPQ_ are indicative of a high photoprotective capacity, whereas high values of Φ_NO_ reflect the inability of a plant to protect itself against damage by excess illumination [61]. Considering both of our samplings were performed at saturating light intensity, the high values of Φ_NO_ and low values of Φ_NPQ_ in the female parents reflect suboptimal capacity of photoprotective reactions, which eventually will lead to photodamage. The high ratio Φ_NPQ_/ Φ_NO_ indicates larger photoprotective capacity in the hybrids.

Seemingly, hybrids resemble more the male parents than the female parents in photosynthetic-related traits, even though chloroplast are mother-inherited. Considering the small size of the plastid genome, a nuclear-plastid cooperative interaction in the development of chloroplast structure and function has been reported [62,63]. Therefore, there is no reason to believe hybrids should resemble more the female than the male parents. Indeed, this study suggests a higher capacity of fitness of the hybrids and male parents to stress conditions. As stated above, the male parents originated from a breeding program specific to the Mediterranean growing regions. This observation reinforces the convenience of carrying out breeding with adapted germplasm and, on the other hand, highlight the potential of hybrids breeding to achieve a combination of desirable traits from two sources in a fast and effective way.

### 4.3. Suitability of MultispeQ as a Tool to Screen Plant Populations for Stress Responses

We have tested the MultispeQ device in a field-based experiment, to characterize a population of genotypes, with the objective to obtain meaningful results in a plant genetics context. We have found some intriguing results only at the level of comparisons between *genotypic sets*. The differences between individual genotypes presented a magnitude not far from the residual resulting from differences between plants belonging to the same genotype. Therefore, more measurements than the four carried out per genotype would be needed to characterize a field plot. The use of two devices, though it increased speed of operation, supposed an additional source of experimental error. An adequate balance of number of devices and plants per plot should be found for a particular experiment, taking into account constraints of time and frequency of measurements.

Photosynthesis is a key informant of the overall fitness of the plant. The use of handheld devices like the one used in our study enables assessing photosynthetic performance at large-scale and affordable price, under field conditions, and opens the possibility to explore traits that were understudied in crop genetics. This type of proximal measurements, as well as the emerging field of solar-induced chlorophyll fluorescence (SiF), which has the additional advantage of being amenable to remote measurement [64,65], and could become useful additions to the breeders’ toolbox. 

## 5. Conclusions

This study proves the suitability of a low-cost fluorimeter to be applied in large-scale field phenotyping aiming at screening genotypic variability in responses to stress. Among the discussed parameters, Φ_NPQ_ was the most sensitive in the detection of differences between genotypic sets, distinguishing between male parents, female parents and hybrids. Male and female parents responded differently, indicating the presence of different mechanisms to cope with water stress. The combination of traits detected in the hybrids seemed favorable. Therefore, improved hybrid cultivars could be built by pyramiding strategic combinations of stress adaptive traits coming from the two parents. 

## Figures and Tables

**Figure 1 sensors-20-01486-f001:**
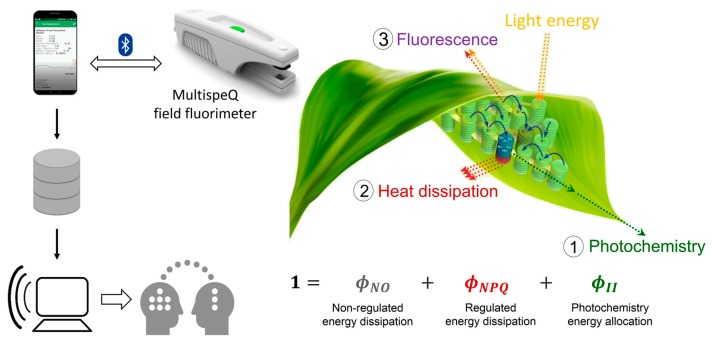
Three competing fates of light energy absorbed by leaf chlorophylls and carotenoids: (**1**) photochemistry, (**2**) heat dissipation, and (**3**) chlorophyll fluorescence. MultispeQ field fluorimeter measures chlorophyll fluorescence in response to light saturating pulses, enabling to estimate the energy proportion allocated towards photochemistry (φ_II_), regulated non-photochemical energy loss (φ_NPQ_), and non-regulated non-photochemical energy loss (φ_NO_). The MultispeQ device is connected via Bluetooth with the Photosynq app installed on a smartphone. The data collected in the field is sent to the cloud, then the users can use analysis tools, discuss and share the data in the web interface.

**Figure 2 sensors-20-01486-f002:**
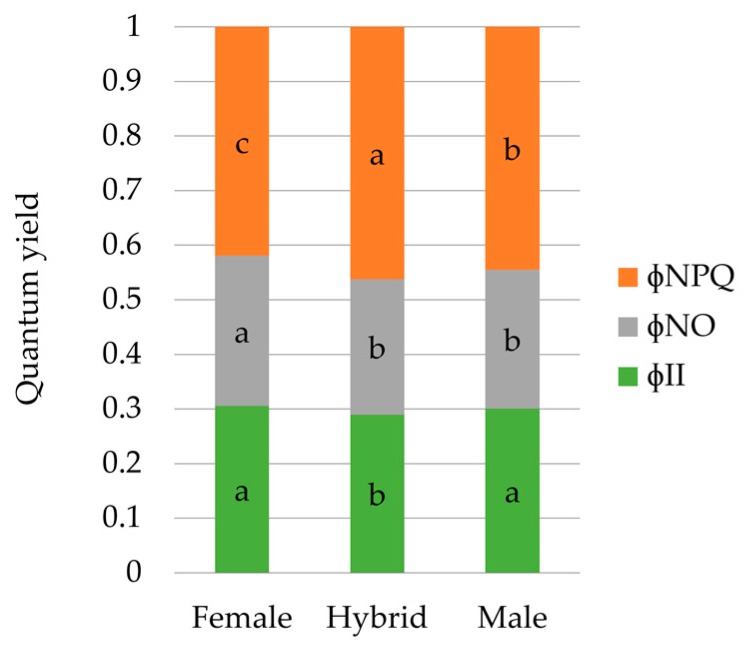
Differences in energy allocation proportions between *genotypic sets* (hybrids of both females combined). Each bar represents the unit, and is split in three parts in terms of the complementary quantum yields of PSII: (**1**) quantum yield of photochemical energy conversion in PSII (φ_II_), (**2**) quantum yield of non-regulated non-photochemical energy loss in PSII (φ_NO_), and (**3**) quantum yield of regulated non-photochemical energy loss in PSII (φ_NPQ_). Each bar represents mean values of energy partition for each genotypic set. Values of the same variable designated with different letters are significantly different at P < 0.05 according to the contrast performed for the overall ANOVA.

**Figure 3 sensors-20-01486-f003:**
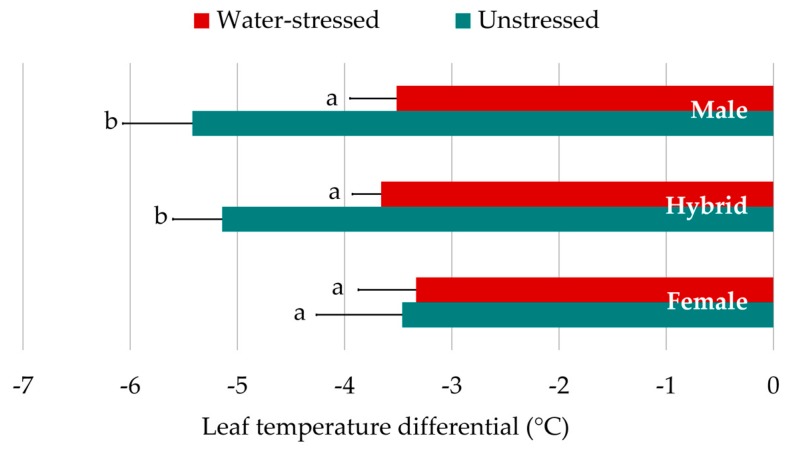
Differences in leaf temperature differential response to water conditions between *genotypic sets* (all hybrids combined). Means ± 95% confidence interval of LTD in °C are represented for each genotypic class, under water-stressed and unstressed conditions. Bars with different letter are significantly different at P < 0.05 according to means separation by LSD.

**Figure 4 sensors-20-01486-f004:**
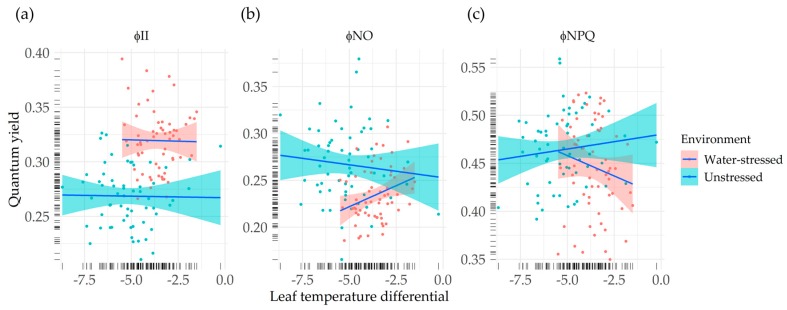
Correlation between quantum yields and leaf temperature differential under *environments*. (**a**) Effective quantum yield of PSII photochemistry vs. LTD; (**b**) quantum yield of non-regulated non-photochemical energy loss in PSII vs. LTD; (**c**) quantum yield of regulated non-photochemical energy loss in PSII vs. LTD. In each panel, genotypic means (dots) and regression lines (lines) are represented for water-stressed (red) and unstressed conditions (blue). Rugs of lines in the axes represent the distribution of the genotypic means.

**Table 1 sensors-20-01486-t001:** Climate data recorded at the two sampling days. Daily mean photosynthetically active radiation (PAR), daily mean temperature (Tm), precipitation accumulated during the three weeks prior to the sampling date (P), daily mean solar irradiance (R), and daily mean relative humidity (RH).

Environment	PAR (*μ*mol m^−2^ s^−1^)*	Tm (°C)*	P (mm)*	R (W m^−2^)*	RH (%)*
Water-stressed	1350	13.8	9.6	260.4	64.7
Unstressed	1404	17.4	90.2	285.8	75.6

* PAR is the mean of PAR values recorded by MultispeQ during all measurements performed each sampling date. Tm, R and RH are daily means recorded by the meteorological station at EEAD. P is the sum of precipitation recorded during the last three weeks prior the sampling date.

**Table 2 sensors-20-01486-t002:** Effects of *genotypic set*, *environment*, device, *genotypic set* by *environment* interaction, and contrasts on variables assessed using MultispeQ. The values under each variable’s heading correspond to mean squares.

Source of Variation	df	RC	LTD	Φ_II_	Φ_NO_	Φ_NPQ_	NPQ_t_	qL	qP
Sqrt (PAR)	1	1655 ***	3.5	3.7 × 10^−1^***	9.2 × 10^−3^	2.6 × 10^−1^***	3.6	0.341 ***	0.661 ***
Genotypic sets (GS)	3	323 **	26.1 *	5.6 × 10^−3^*	2.3 × 10^−2^***	2.3 × 10^−2^**	5.5 **	0.026 ***	0.015 **
F_1_ *vs* Parents	1	67	8.8	5.5 × 10^−3^·	2.4 × 10^−2^*	5.3 × 10^−2^**	10.4 **	0.017 *	0.000
F_1_ *vs* Females	1	906 ***	64.9 **	8.3 × 10^−6^	4.4 × 10^−2^***	4.6 × 10^−2^**	10.6 **	0.049 ***	0.020 *
F_1_ *vs* Males	1	30	0.3	6.9 × 10^−3^*	5.3 × 10^−3^	2.4 × 10^−2^*	4.3·	0.002	0.003
Females *vs* Males	1	958 ***	65.7 **	2.0 × 10^−4^	3.8 × 10^−2^**	3.3 × 10^−2^*	7.9 *	0.044 **	0.023 *
F_1_ (FemA) *vs* F_1_ (FemB)	1	11	12.0	9.9 × 10^−3^*	2.1 × 10^−2^*	2.0 × 10^−3^	2.3	0.027 **	0.021 *
Genotypes within GS	57	53 **	6.7 *	1.7 × 10^−3^	3.6 × 10^−3^	5.0 × 10^−3^	1.1	0.004	0.003
Environment (ENV)	1	54,009 ***	317.3 ***	2.7 × 10^−1^***	1.3 × 10^−1^***	2.3 × 10^−2^·	14.6 **	0.049 ***	0.636 ***
Device	1	1363 ***	359.6 ***	1.0 × 10^−1^***	1.6 × 10^−1^***	5.2 × 10^−1^***	80.8***	0.069***	0.000
Genotypic set*ENV	3	86·	7.43	2.9 × 10^−3^	1.1 × 10^−2^	3.0 × 10^−3^	0.6	0.013	0.010
F_1_ *vs* Parents*ENV	1	170 *	0.3	3.0 × 10^−4^	3.4 × 10^−3^	1.7 × 10^−3^	0.1	0.000	0.000
F_1_ *vs* Females*ENV	1	95	19.0·	2.7 × 10^−5^	1.0 × 10^−4^	2.0 × 10^−4^	0.2	0.001	0.002
F_1_ *vs* Males*ENV	1	102·	2.3	5.0 × 10^−4^	3.9 × 10^−3^	1.6 × 10^−3^	0.0	0.001	0.000
Females *vs* Males*ENV	1	59	21.4·	1.0 × 10^−4^	1.0 × 10^−5^	4.0 × 10^−5^	0.1	0.002	0.002
F_1_ (FemA) *vs* F_1_ (FemB)*ENV	1	75	0.0	8.1 × 10^−3^*	3.1 × 10^−2^*	7.2 × 10^−3^	1.5	0.035 *	0.027 *
Genotypes within GS*ENV	57	35	5.7	1.7 × 10^−3^	6.1 × 10^−3^	7.5 × 10^−3^	1.7	0.006 *	0.004·
Residuals	412	31	4.5	1.9 × 10^−3^	4.8 × 10^−3^	6.6 × 10^−3^	1.5	0.004	0.003
**Source of variation**	**df**	**LEF**	**RFd**	**F_m_′**	**F_0_′**	**F_s_**	**F_v_′/F_m_′**		
sqrt(PAR)	1	164,327 ***	1.753 ***	2.2 × 10^8^ ***	1.3 × 10^7^ ***	3.0 × 10^6^	1.78 × 10^−2^·		
Genotypic sets (GS)	3	2329 *	0.020*	8.1 × 10^6^	8.7 × 10^5^	4.9 × 10^6^	2.26 × 10^−2^ **		
F1 vs Parents	1	2271·	0.022·	2.1 × 10^7^	6.7 × 10^4^	5.5 × 10^6^	3.97 × 10^−2^**		
F1 vs Females	1	12	0.000	4.0 × 10^6^	2.3 × 10^6^·	2.1 × 10^6^	3.97 × 10^−2^**		
F1 vs Males	1	3149 *	0.026·	1.8 × 10^7^	2.0 × 10^5^	3.8 × 10^6^	4.67 × 10^−2^**		
Females vs Males	1	216	0.000	2.0 × 10^5^	2.5 × 10^6^·	1.1 × 10^6^	1.46 × 10^−2^·		
F1 (FemA) vs F1 (FemB)	1	3792 *	0.033 *	3.6 × 10^6^	8.0 × 10^3^	9.0 × 10^6^	3.62 × 10^−2^**		
Genotypes within GS	57	633	0.007	1.0 × 10^7^	7.3 × 10^5^	5.4 × 10^6^	4.08 × 10^−3^		
Environment (ENV)	1	111,993 ***	1.120 ***	1.0 × 10^8^**	2.2 × 10^7^***	7.4 × 10^5^	3.07 × 10^−2^*		
Device	1	34,605 ***	0.494 ***	4.3 × 10^8^***	3.2 × 10^6^*	1.0 × 10^8^***	3.27 × 10^−1^***		
Genotypic set*ENV	3	1519	0.008	1.4 × 10^7^	9.8 × 10^5^	8.7 × 10^6^	4.43 × 10^−3^		
F1 vs Parents*ENV	1	56	0.002	6.6 × 10^6^	1.9 × 10^6^	2.3 × 10^6^	3.94 × 10^−1^***		
F1 vs Females*ENV	1	57	0.000	2.5 × 10^7^	2.5 × 10^6^·	1.1 × 10^7^	3.00 × 10^−4^		
F1 vs Males*ENV	1	153	0.003	2.0 × 10^5^	6.2 × 10^5^	7.0 × 10^3^	5.47 × 10^−6^		
Females vs Males*ENV	1	99	0.000	2.4 × 10^7^	2.0 × 10^6^	1.1 × 10^7^	4.00 × 10^−4^		
F1 (FemA) vs F1 (FemB)*ENV	1	4336 *	0.021·	1.6 × 10^7^	6.9 × 10^2^	1.5 × 10^7^	4.09 × 10^−5^		
Genotypes within GS*ENV	57	739	0.006	1.5 × 10^7^	8.6 × 10^5^·	7.7 × 10^6^·	6.68 × 10^−3^		
Residuals	412	768	0.007	1.2 × 10^7^	6.6 × 10^5^	6.1 × 10^6^	5.48 × 10^−3^		

df, degrees of freedom; · *P* < 0.1 * *P* < 0.05 ** *P* <0.01 *** *P* <0.001.

**Table 3 sensors-20-01486-t003:** Mean ± 95% confidence intervals of MultispeQ-derived traits, sorted by *environment* (unstressed vs. water-stressed) and genotypic set (female, hybrid, and male). *Environment* means are averaged for all genotypes. Genotypic means are averaged for the two environments. Asterisks indicate significantly different environment means at P < 0.05 according to overall ANOVA. Genotypic set means with different letter are significantly different at P < 0.05 according to means separation by LSD.

	Environment Effect			Genotypic Effect		
Trait	Unstressed	Water-Stressed		Female	Hybrid	Male
**RC**	63.5 ± 0.82	43.0 ± 0.64	***	56.0 ± 2.49^a^	53.1 ± 1.38^b^	52.4 ± 1.81^b^
**LTD**	−5.02 ± 0.35	−3.58 ± 0.21	***	−3.40 ± 0.47^a^	−4.40 ± 0.28^b^	−4.47 ± 0.41^b^
**φ_II_**	0.270 ± 0.006	0.321 ± 0.006	***	0.306 ± 0.016^a^	0.290 ± 0.006^b^	0.301 ± 0.009^a^
**Φ_NO_**	0.270 ± 0.010	0.236 ± 0.008	***	0.275 ± 0.017^a^	0.248 ± 0.009^b^	0.254 ± 0.011^b^
**Φ_NPQ_**	0.461 ± 0.009	0.443 ± 0.013	ns	0.419 ± 0.019^c^	0.462 ± 0.010^a^	0.445 ± 0.013^b^
**NPQ_t_**	1.93 ± 0.10	2.27 ± 0.19	**	1.70 ± 0.22^b^	2.22 ± 0.16^a^	2.02 ± 0.18^ab^
**qL**	0.226 ± 0.009	0.295 ± 0.008	***	0.246 ± 0.020^a^	0.263 ± 0.009^a^	0.262 ± 0.012^a^
**qP**	0.442 ± 0.010	0.522 ± 0.006	***	0.483 ± 0.023^a^	0.480 ± 0.009^a^	0.490 ± 0.012^a^
**LEF**	168.97 ± 3.23	192.51 ± 4.93	***	172.2 ± 8.9^b^	180.7 ± 4.1^a^	184.3 ± 5.6^a^
**RFd**	0.375 ± 0.011	0.481 ± 0.014	***	0.454 ± 0.034^a^	0.417 ± 0.013^b^	0.439 ± 0.017^a^
**F_m_′**	11657 ± 421	12656 ± 457	**	12635 ± 1025^a^	11930 ± 413^a^	12406 ± 537^a^
**F_0_′**	4139 ± 108	4576 ± 93	***	4247 ± 266^a^	4352 ± 97^a^	4413 ± 120^a^
**F_s_**	8541 ± 334	8457 ± 269	ns	8684 ± 686^a^	8407 ± 288^a^	8605 ± 357^a^
**F_v_′/F_m_′**	0.631 ± 0.008	0.617 ± 0.011	*	0.651 ± 0.016^a^	0.616 ± 0.009^b^	0.628 ± 0.011^b^
**n**	268	268		64	312	160

n, number of replicates; ns, not significant; * *P* < 0.05 ** *P* < 0.01 *** *P* < 0.001.

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
