# Peer review of "Rapid On-Site Phenotyping via Field Fluorimeter Detects Differences in Photosynthetic Performance in a Hybrid—Parent Barley Germplasm Set"

_sensors, 2020, doi:10.3390/s20051486_

Round 1

Reviewer 1 Report

The manuscript by  Fernández-Calleja et al. : “Rapid on site phenotyping via field fluorimeter 2 detects differences in photosynthetic performance in 3 a hybrid-parent barley germplasm set” deals with testing of instrument MultispeQ  for plant phenotyping in field conditions.  The instrument MultispeQ is a very promising tool for breeders, agronomists and plant physiologists as it combines three measurement approaches: chlorophyll fluorescence, spectral transmittance and leaf temperature, and thus potentially allows to detect a number of abiotic (and probably also biotic) stresses. The manuscript is well organized, clearly written, with few inaccuracies that I mention below. Most of them could be easily corrected.    However, the major problem of this study  I see in the methodology. It is a pity because this study could be beneficial and important for further development of plant phenotyping approaches based on the combination of several indirect methods. The main problems of the study that make difficult the interpretation of the results are:

Indirect measurements of physiological status based on chlorophyll fluorescence, spectral transmittance (chlorophyll content) and leaf temperature difference (stomatal conductance) are not compared in the study with the direct indicators of plant response to drought such as relative water content in leaves, water potential, biomass production, grain yield, photosynthetic parameters, transpiration rate etc. I still hope that authors have at least one of these parameters available, and the data will be then publishable. The drought stress and unstressed conditions took place at different time and growth stage. The differences in time could be caused not only by water availability but also by temperature, changes in nutrient availability (dilution of nitrogen with the development), senescence in older leaves, different morphology of later developed leaves (leaf thickness), disease infection etc. This is not so critical as the previous point, and hopefully, this issue can be reduced by improved statistical analysis. Each of genotype, i.e. males, females and hybrids was cultivated only in one plot/replicate. Absence of real replicates does not allow to statistically evaluate differences between genotypes. The use of genotypes within group males, females and hybrids as replicates could be acceptable for regression analysis with the direct indicators of response to drought stress mentioned in point (i).

Mainly due to these methodological problems I recommend rating “Major revisions”,  and publishing the manuscript only if some direct parameters of water stress response are added (such as relative water content, water potential, biomass production, grain yield, stomatal conductance etc.) .

I have also several other requirements for corrections that are, however, of less importance and generally relatively easy to be met:

Abstract

90% of the abstract consists of an introduction and methodology. It is virtually unclear what results were achieved and what the conclusions are.  Please shorten the introduction part and include the clear message emerging from your results.

L. 38-40 … is key to construct accurate statistical models. It is not very clear what you want to to communicate. Could you please explain how the phenotyping is important to construct statistical models

L. 47-49 Field‐based plant phenomics enables trait‐based crop breeding (physiological breeding ….. both field and greenhouse phenotyping have a goal – trait based crop breeding.  If I understood correctly you are trying to emphasize the role of phenotyping based on physiological traits which is also not limited to field conditions. Why you don't start with …Phenotyping based on physiological traits (physiological breeding …

L. 66-70 this statement is not correct. Both light acclimated and dark acclimated leaves are used for measurement of chlorophyll fluorescence, but the different parameters with different biophysical meaning are derived. Also the steady state measurement of chlorophyll fluorescence is available with other instruments

L. 104 average solar radiation of 187.2 W m-2 you mean global solar radiation (irradiance) ?

Could you please briefly specify the soil type and texture?

L. 123-126 Why different numbers of males and hybrids were used for measurement after drought stress and after rain episode? Could it affect the results? Why only the genotypes included in both measurements were not used for analysis. Please explain.

L. 129-130 4 measurements per genotype would be sufficient when the plots were replicated but only 4 measurements per genotypes can be a source of high variability and low significance of results.

Table 1  in this table is rather confusing which data are daily means (global radiation ?),   or actual data during the measurement (probably PAR?), or 3 week sum (precipitation ..and some others?). Please try to indicate it already in the table. This table needs to be commented also in the text.  No reference of Table 1 is in the text.

L. 175-177 The yield induced by downregulatory processes (φNPQ), yield for other energy losses (φNO), and quenching due to non-photochemical dissipation of absorbed light energy (NPQt) were estimated according to [30]. Could you please explain how you calculated the φNO and φNPQ parameters? For this calculation, you need Fm which requires dark acclimation for min 15 minutes, but this was probably not done in the field.  

L. 181 Relative chlorophyll content (RC) was calculated as 100 × log(transmittance@940/transmittance@650), which accounts for the thickness of the leaf [11]. This sentence is confusing. Transmittance measurement is affected by leaf thickness but the major factor is chlorophyll content and if you express the chlorophyll content per area unit it is clear that it is integral of relative content of chlorophyll and leaf thickness. So please change the sentence e.g. Relative chlorophyll content (RC) was calculated as 100 × log(transmittance@940/transmittance@650)  which is expressed per area unit and it is the value integrating the chlorophyll content per unit mass and leaf thickness.

L. 185-197 I am not sure about the correct use of ANOVA for such type of experiment. This is rather an observational type of study (not experiment with true replications) and it could be expected that the predictors are correlated probably is better to use ANCOVA. It could for example filter out the effect of PAR and maybe also the differences between genotypes. But I leave this decision to the authors.   

Fig. 3 LTD under water stress indicates only mild water stress.  Under severe water stress, the LTD  is between -1 and 0 and in combination with high radiation it can be above 0.  This is also probably the reason for relatively low correlations between LTD and fluorescence parameters. Although the drought was clearly defined by almost no precipitation for 3 weeks,  the level of drought stress will be better characterized by soil moisture measurement.  

Author Response

The responses are attached in a word file. 

Reviewer 2 Report

After writing a lengthy praise and critique of the manuscript here something has gone wrong with the review platform and everything has been erased.

In short summary, the manuscript is well-prepared and the research thoughtful. I recommend a minor editorial review of details in the English grammar with a few small errors here and there. 

The ideas in the very short sentence on line 47 may be of more interest than just one sentence and may be further expanded with other more concrete examples of the trait-based phenotyping movement. This may also help to transition from phenotyping to photosynthesis in the Intro.

Further introduction to hybrid development and later on discussion on other related benefits of barley hybrids from other studies apart from the specific details of note here may also be of interest.

ln 89 no comma is needed after "tool"

How was the climate data from Table 1 acquired?

"Crop husbandry followed local practices" could also use some minor expansion towards study reproducibility.

Please expand on the details of the "associated macro" identified on ln. 142 and 166.

Are other studies using the MultiSpeQ or PhotosynQ already published?

Any potential for tying into the hot topic of SIF? Would this study make SIF more relevant for phenotyping?

I am left with the feeling that the Conclusions could be somewhat more developed.

Author Response

(The authors gave the same response as above.)

Reviewer 3 Report

In my opinion it I a well written manuscript with very interesting and useful information. However, you should provide more details to be accepted.

Introduction: I appreciate to include a revision about how (sensors / platforms) these activities are performed.

Materials:

Line 101, include CRS.

Line 130, please details how many plots.

Line 135, any reason about your sample size? Which sampling method did you use?

Line 138, please, include an image with the built mask.

Line 139, You recalibrated your devices. How did you validate this activity?

Line 188. Did you analyze preconditions to use ANOVA?, please provide more information in this section and include test in result section.

Table 2. You wrote in caption about figures, which figures?

Section 3.2 If you create a subsection it has to be important and provide information. In this case , you wrote only 3 lines...

Line 265. How did you measure air temperature? Your measurements have to be accurate and precise because we are using a small differential. What is accuracy and precision of these devices? Are they stable? What about drift?

Conclusion: you have to rewrite it, and conclude. In this version, I read just only a very short summarize of your document.

Do you think it can be complementary information to other sources?

Author Response

(The authors gave the same response as above.)

Round 2

Reviewer 1 Report

The authors have eliminated major shortcomings and although I still have minor reservations about the experimental design or NPQ calculation methodology (maximum quantum yield of PSII Fv/Fm is also affected by severe stress and the use of the theoretical value of 0.83 may not always be correct) in the case of this study it plays minor role (mild stress) and thus I recommend to accept the manuscript for publication.

Author Response

(The authors gave the same response as above.)

Reviewer 3 Report

You change coordinates but you have not include coordinate reference system.

In my opinion, a conclusion section has to be in a scientific manuscript. Please, write it.

Author Response

(The authors gave the same response as above.)
